# Composting Process and Gas Emissions during Food Waste Composting under the Effect of Different Additives

Hyun Young Hwang [ID], Seong Heon Kim, Jaehong Shim and Seong Jin Park *

Rural Development Administration, National Institute of Agricultural Sciences, Wanju 55365, Korea; hyhwang2325@korea.kr (H.Y.H.); ksh4054@korea.kr (S.H.K.); jayshim@korea.kr (J.S.)
* Correspondence: archha98@korea.kr; Tel.: +82-63-238-2452

**Abstract:** This study investigated the effects of adding mature compost (MC) and vermicompost (VC) on controlling gas emissions and compost quality during food waste (FW) composting. In addition to a control treatment (only food waste), four treatments were designed to mix the initial FW with varying rates of MC and VC (5.0% and 7.5%). The composting process was monitored for 84 days. Results indicate that the addition of MC and VC resulted in higher temperature, prolonged the thermophilic stage and reduced $NH_3$ and greenhouse gas (GHG) emissions. Compared to the control, the loss of $NH_3$-N was decreased by 29–69%, and the global warming impact was also mitigated by 49–61%. The largest reductions in $NH_3$ and global warming potential (GWP) were found for 7.5% VC and 5% MC, respectively. The treatments with additives more rapidly achieved the required maturity value. This research suggests that the addition of 7.5% MC and VC is suitable for food waste composting.

**Keywords:** ammonia; greenhouse gas; mature compost; vermicompost; kitchen waste

## 1. Introduction

Food wastes (FW) are produced throughout the food supply chain. Approximately 14,000 tons of FW are produced daily, and over five million tons need to be incinerated or disposed of landfills in Korea [1]. These disposal methods have caused environmental problems. Public attention toward FW treatment has increased with the growing interest of environmental issues.

Composting is a proper valorization strategy for FW involving the transformation of organic wastes into organic amendments [2,3]. The land application of compost can increase the level of soil organic carbon (SOC) stocks, improve the soil structure, and reduce the use of chemical fertilizer. However, one of the most important issues related to FW composting is associated with ammonia ($NH_3$) and greenhouse gases (GHGs) such as methane ($CH_4$) and nitrous oxide ($N_2O$), which can contribute to global warming and stratospheric ozone depletion. The global warming potential (GWP) of $CH_4$ and $N_2O$ is 25 and 298 times that of carbon dioxide ($CO_2$), respectively [4]. $NH_3$ is the largest component of gaseous emissions during composting, and a precursor of small particulate matter ($PM_{2.5}$). Furthermore, the loss of $NH_3$ reduces the nutrient quality of compost.

Several studies investigated various types of additives to decrease the loss of gases, such as coal ash [5], zeolite [6,7], vermiculite [8] and biochar [9,10] for the composting of various organic waste types. In particular, the effects of mature compost (MC) were compared when mixed and when covered [11], and also under different mixing rates of MC [12]. However, the optimal mixing rate has not yet been discussed. Although data in the literature indicate that vermicompost (VC) contains humified, stable organic compounds [13], the application possibility of VC as an additive has not

been well tested during food waste composting. Further studies of compost additives are required to provide proper strategies to mitigate the loss of gases in FW composting.

The aim of this study was to investigate the potentials of using MC and VC as additives to reduce GHGs and $NH_3$ emissions during food waste composting. Their feasibility was explored with different quantities of two additives. The optimal additive material and its amount were identified and its nutrient content and maturity were also determined to evaluate its quality for compost production.

## 2. Materials and Methods

### 2.1. Composting Materials

For the experiment, food waste was obtained from an enclosed, local municipal waste collection station (Damyang, Korea). Its effective composition included vegetables, fruits, staple food, and meat. The food waste (30 kg wet weight) was mixed with 30% (*w/w*) sawdust (9 kg wet weight) to control the C/N ratio between 20 and 30 and the initial moisture content to about 60%. As shown in Table 1, food waste has high electrical conductivity due to the sodium content and relatively great nitrogen concentration. Mature compost (MC) and vermicompost (VC) were then incorporated into the initial composting stock with varying rates of 5% and 7.5% of the fresh weight of the compost pile. The food waste compost without MC or VC was taken as the control. The five treatments were labeled as follows: 5% MC, 7.5% MC, 5% VC, 7.5% VC, and control. Mature compost from a previous composting test and VC sold in market were used in this study. Detailed characteristics of the food waste, mature compost, vermicompost, and sawdust are shown in Table 1.

**Table 1.** Properties of the composting materials (mean value ± standard deviation from triplicate measurements).

| Parameter | Food Waste | Mature Compost | Vermicompost | Sawdust |
|---|---|---|---|---|
| Total C (%) | 44.9 ± 0.1 | 39.3 ± 0.3 | 8.1 ± 0.2 | 48.7 ± 0.3 |
| Total N (%) | 5.1 ± 0.03 | 1.1 ± 0.05 | 0.79 ± 0.07 | 0.12 ± 0.00 |
| C/N ratio | 8.8 ± 0.1 | 35.1 ± 1.7 | 10.2 ± 0.8 | 400 ± 17.7 |
| DOC (g kg$^{-1}$)[1] | 107 ± 13 | 6.3 ± 0.22 | 1.2 ± 0.05 | 5.6 ± 0.08 |
| DON (g kg$^{-1}$)[1] | 16.4 ± 2.1 | 2.8 ± 0.04 | 2.1 ± 0.08 | 0.27 ± 0.004 |
| HWEC (g kg$^{-1}$)[1] | 25.2 ± 3.0 | 4.2 ± 0.12 | 1.7 ± 0.09 | 3.8 ± 0.13 |
| HWEN (g kg$^{-1}$)[1] | 3.2 ± 0.36 | 1.1 ± 0.02 | 0.32 ± 0.002 | 0.11 ± 0.003 |
| EC (dS m$^{-1}$)[1] | 7.0 ± 0.1 | 2.6 ± 0.07 | 2.2 ± 0.03 | 0.25 ± 0.02 |
| pH | 4.9 ± 0.1 | 7.4 ± 0.1 | 7.0 ± 0.1 | 5.3 ± 0.1 |

[1] DOC and DON: dissolved organic carbon and nitrogen, HWEC and HWEN: hot water extractable carbon and nitrogen, EC: electrical conductivity.

### 2.2. Experiment Design

The conventional static chamber method was chosen to perform this composting experiment for the spring-summer season. Five 62 L plastic boxes covered with 5 cm thick polystyrene were used to prevent heat loss. The treatments were not replicated because the composting scale ensured experimental reproducibility, as evidenced in other studies [14–16]. The temperature of all compost piles was continuously monitored using a data logger (EM50 Data logger, Washington, DC, USA).

### 2.3. Sampling and Analytical Methods

Greenhous gases (GHGs) such as carbon dioxide ($CO_2$), methane ($CH_4$) and nitrous oxide ($N_2O$) were collected on days 2, 9, 16, 23, 30, 37, 44, 51, 61, 71 and 84 using the closed chamber method. The concentrations of GHGs were quantified using gas chromatography with a methanizer (Shimadzu, GC-2010, Tokyo, Japan). $NH_3$ was captured using 0.1 mol L$^{-1}$ sulfuric acid. The ammonia concentration in the acid was analyzed using Auto Analyzer 3 (Bran Luebbe, Hamburg, Germany). Details related to the sampling and measurement of gases are clearly explained in a previous study [16].

During the 84 days of composting, the compost piles were turned over and thoroughly mixed every week at the fixed time of 09:00. After turning, compost piles were collected by a sampling core at three points. The collected samples were divided into two parts; one portion was preserved at 4 °C before analysis, and the other portion was air-dried, ground and sieved with a 2 mm mesh.

Total organic carbon and nitrogen contents were quantified using an elemental analyzer (CHNS-932 Analyzer, Leco, Calagry, AB, Canada). The compost samples were shaken with distilled water (1:20 *w/w*) for 2 h, and pH and electrical conductivity (EC), an index of salinity of food waste, were measured by pH and EC meter (Orion 3star, Thermo Electron Corporation, Waltham, MA, USA). The extract was filtered through a 0.45 µm membrane filter to analyze the water extractable carbon and nitrogen concentration; subsequently, hot water extractable C and N were extracted using distilled water at 80 °C for 16 h, as they are relatively labile organic compounds. Concentration was determined using a TOC-5050A analyzer (Shimadzu Corporation, Tokyo, Japan). Fresh samples were extracted using 2 M potassium chloride (1:10, *w/v*) to determine the concentration of ammonium ($NH_4^+$) and nitrate ($NO_3^-$) using a segmented flow analyzer (Bran Luebbe, Hamburg, Germany). The germination index (GI) was used to assess the phytotoxicity of the compost [16,17].

### 2.4. Statistical Analysis

Principle component analysis was performed using R software to determine the relationship between compost characteristics and gas emissions for all treatments during composting. For each treatment, the correlation among composting properties was calculated and illustrated through a heat map using R software at a statistical significance level of 0.05.

## 3. Results and Discussion

### 3.1. Changes in Gas Emissions during Composting

The dominant emissions of $NH_3$ occurred during the initial thermophilic phase. From the 30th day to the end of composting, the $NH_3$ volatilization rate showed a declining trend before lastly tending to the baseline (Figure 1a). The 7.5% MC, 5% VC and 7.5% VC peaked at day 9 (59, 69 and 55 mg $kg^{-1}$ $h^{-1}$), whereas the control and 5% MC peaked later at day 23 (75 and 58 mg $kg^{-1}$ $h^{-1}$). This was possibly due to the conversion of ammonium ($NH_4^+$) to $NH_3$, which was attributed to the rapid biodegradation of organic nitrogen (N) to inorganic N. $NH_4^+$ can volatilize becoming $NH_3$ under high temperature and high pH conditions. The trends of $NH_3$ production correlated with temperature, albeit with a delayed and slower decrease; this result agrees with the results of He et al. [8] and Zhang et al. [18], who investigated food waste composting using vermiculite and volatile sulfur compounds, respectively.

The use of MC and VC significantly reduced $NH_3$ emissions during composting. Specifically, the 7.5% MC and 7.5% VC treatments effectively reduced $NH_3$ emissions in the thermophilic phase (Figure 1a). A lower reduction was observed in the 5% MC and VC treatments. Both 7.5% treatments showed a negligible level from 23rd day onward; however, the 5% treatments experienced dramatic drops on days 30 and 37. This result agrees with that reported by Yang et al. [12] who found that adding mature compost could decrease initial $NH_3$ production via adsorption and subsequent nitrification.

$N_2O$ emissions were effectively reduced by the addition of MC and VC throughout the composting period (Figure 1b). Methanotrophic bacteria can oxidize $NH_3$ to $N_2O$ as reported in many studies [19,20]. This finding could also be confirmed by the results of this study, whereby the highest $N_2O$ emissions were found in the control, which produced the greatest $NH_3$ volatilization rate. Since high temperature (over 40 °C) can hinder the activity of nitrifiers, the considerable $N_2O$ emissions in the thermophilic stage were possibly due to $NH_4^+$ oxidization by methanotrophs [21,22].

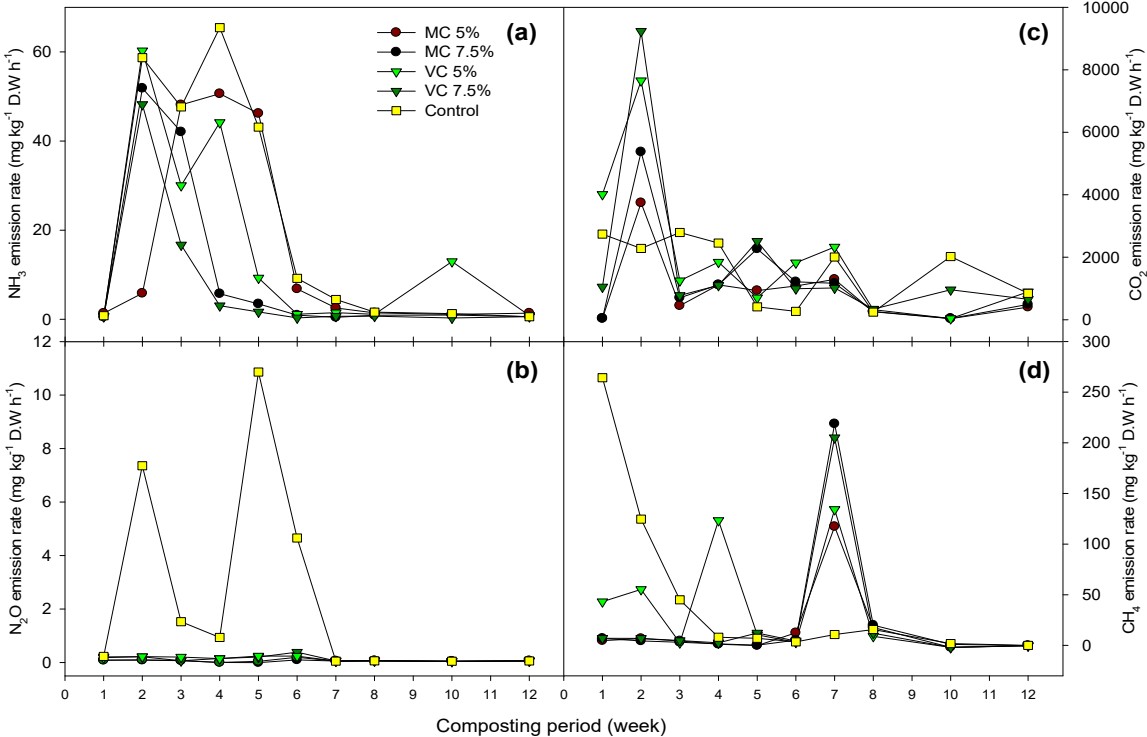

**Figure 1.** (**a**) Ammonia, (**b**) carbon dioxide, (**c**) nitrous oxide, and (**d**) methane emissions from different treatments during composting. MC 5%: kitchen waste+sawdust+5% mature compost; MC 7.5%: kitchen waste+sawdust+7.5% mature compost; VC 5%: kitchen waste+sawdust+5% vermicompost; VC 7.5%: kitchen waste+sawdust+7.5% vermicompost, Control: kitchen waste+sawdust.

$CH_4$ emissions showed a considerable difference among treatments (Figure 1d). Although aeration was provided, anaerobic zones could still have been present in the composting pile due to methanogenic activity leading to rapid $O_2$ consumption [23]. For the control treatment, $CH_4$ emissions significantly increased within 10 days and then dramatically decreased to zero, whereas the 5% MC, 7.5% MC and 7.5% VC treatments showed peak emissions on week 7 of composting. $CH_4$ emission peaks of 264 mg kg$^{-1}$ h$^{-1}$ (Week 1), 218 mg kg$^{-1}$ h$^{-1}$ (Week 7), 205 mg kg$^{-1}$ h$^{-1}$ (Week 7), 134 mg kg$^{-1}$ h$^{-1}$ (Week 7) and 117 mg kg$^{-1}$ h$^{-1}$ (Week 7) were recorded for the control, 7.5% MC, 7.5% VC, 5% VC and 5% MC treatments, respectively. Since MC and VC can enhance the air space and adjust the moisture level, with other bulking agents, initial $CH_4$ production could have been reduced. Interestingly, the 5% VC emitted slightly more $CH_4$ than the other additive treatments did in the initial phase, which could have been due to it having the highest $CO_2$ production. A high $CO_2$ concentration would be favorable for $CO_2$-dependent methanogenesis in the compost pile, as confirmed by the positive relationship between the $CH_4$ and $CO_2$ emission patterns (Figure 4).

Carbon dioxide emission is an indicator of microbial activity and the degradation of organic matter (OM) [24,25]. Overall, the $CO_2$ emissions of each treatment were greater in the initial thermophilic phase when the decomposition of organic matter was faster. $CO_2$ emission peaks of 9.23 g kg$^{-1}$ h$^{-1}$ (Week 2), 7.65 g kg$^{-1}$ h$^{-1}$ (Week 2), 5.37 g kg$^{-1}$ h$^{-1}$ (Week 2), 3.74 g kg$^{-1}$ h$^{-1}$ (Week 2) and 2.74 g kg$^{-1}$ h$^{-1}$ (Week 1) were recorded for the 7.5% VC, 5% VC, 7.5% MC, 5% MC and control treatments, respectively (Figure 1c).

### 3.2. Changes in Temperature and pH during Composting

A composting period is generally divided into thermophilic, mesophilic, and curing phases with the mineralization of organic materials. Temperature changes recorded for all treatments are illustrated in Figure 2a.

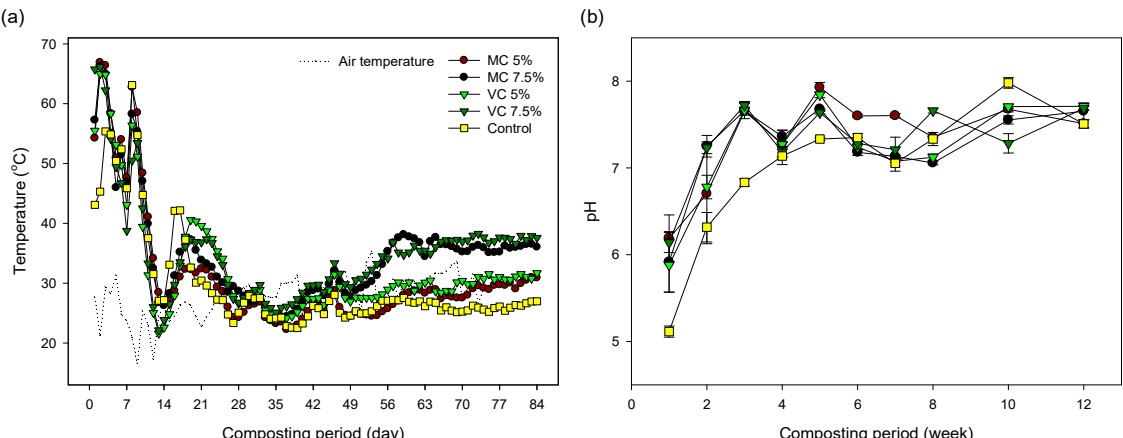

**Figure 2.** Changes in (**a**) temperature and (**b**) pH during composting. MC 5%: kitchen waste +5% mature compost; MC 5%: kitchen waste+sawdust+5% mature compost; MC 7.5%: kitchen waste+sawdust+7.5% mature compost; VC 5%: kitchen waste+sawdust+5% vermicompost; VC 7.5%: kitchen waste+sawdust+7.5% vermicompost, Control: kitchen waste+sawdust.

It is generally accepted that 50 °C is the threshold between the mesophilic and thermophilic phases of composting [26,27]. Except for the control, the temperature of all treatments rapidly increased and reached the thermophilic phase (>50 °C) on the first day of composting before remaining above this level at least for 12 days (Figure 2a), indicating the active biodegradation of organic matter. The treatments containing mature compost and vermicompost attained the highest temperature on day 2 (65–67 °C), while the control treatment reached the highest temperature on day 8 of composting (63 °C). This was mainly due to the intensive degradation of OM. With the steady depletion of biodegradable organic matters, all treatments showed a decline in temperature. However, the temperature of the 7.5% MC and 7.5% VC treatments gradually increased to 40 °C, which was maintained during the curing phase (Figure 2a), possibly due to the additives. Other treatments trended toward ambient temperature on day 29, implying the stabilization of decomposition.

The degradation of organic matter was correlated with an increase in the pH of the composting pile (Figure 2b), which in turn favored $NH_3$ volatilization. The pH of all treatments followed a similar trend, with a rapid increase within 3 weeks, followed by the maintenance of an alkaline state. This occurred because the high temperature led to the volatilization of organic acids, thus increasing the pH value in all treatments. Concurrently, a large amount of $NH_3$ was produced during this stage (Figure 1a). At the end of composting, the pH of all treatments ranged from 7.0 to 8.5, which reflected safe compost standards [28,29]. Principal component analysis (PCA) also confirmed that the pH had a positive relationship with the germination index (Figure 4).

### 3.3. Compost Chemical Characteristics

A faster increase in the germination index (GI) was observed for the additive treatments (Figure 3c). The GI value increased to above 80%, indicating maturity [16,30], in week 5 to 6 and 6 to 7 for the MC and VC treatments, respectively, in comparison to that on week 7 for the control. Mature compost can accelerate the composting process, as a result of the succession of the microbial community [31].

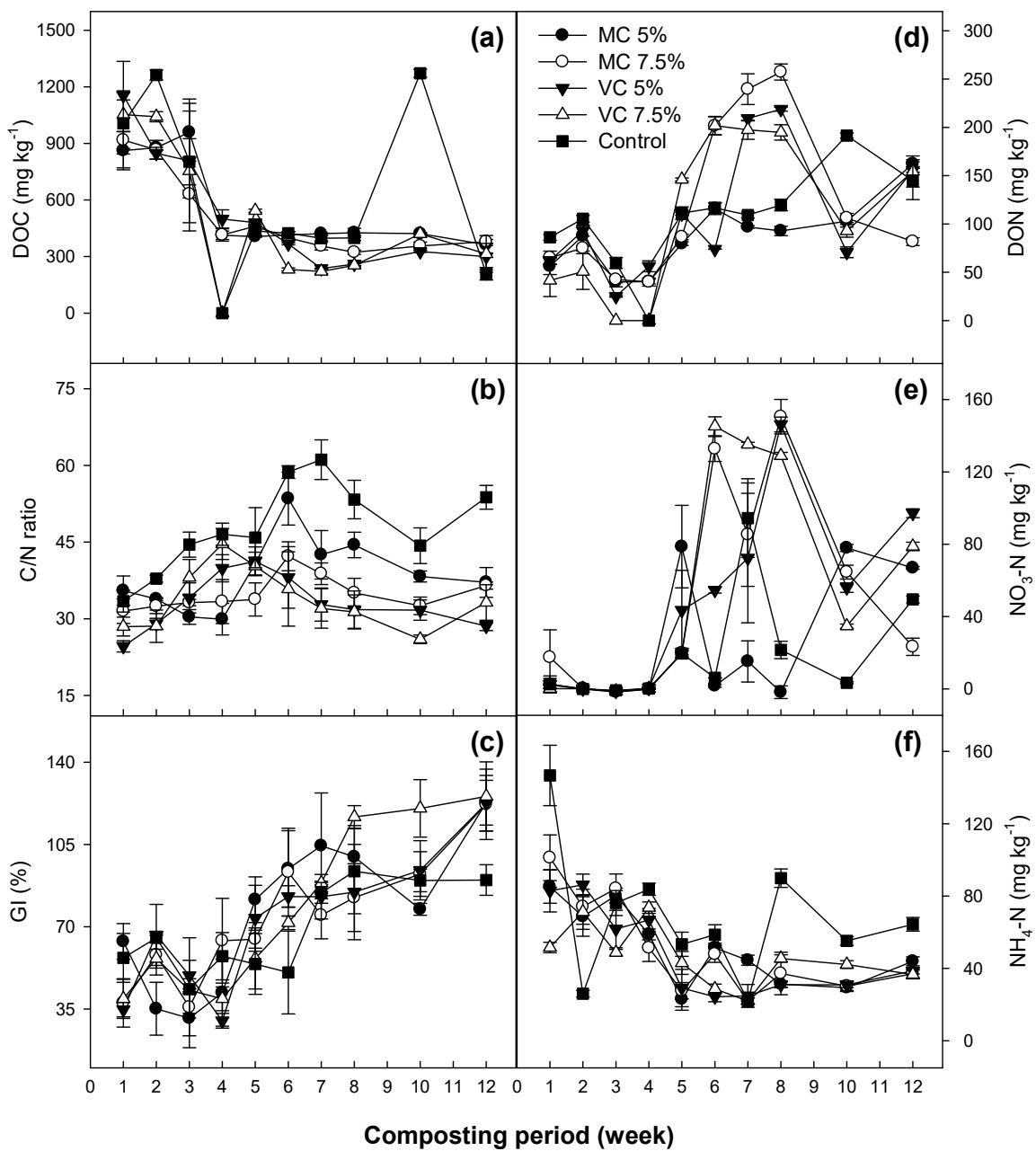

**Figure 3.** (**a**) Dissolved organic carbon, (**b**) dissolved organic nitrogen, (**c**) C/N ratio, (**d**) NO$_3$-N, (**e**) germination index, and (**f**) NH$_4^+$-N content during composting. Error bars represent standard deviations of triplicate measurements. MC 5%: kitchen waste+sawdust+5% mature compost; MC 7.5%: kitchen waste+sawdust+7.5% mature compost; VC 5%: kitchen waste+sawdust+5% vermicompost; VC 7.5%: kitchen waste+sawdust+7.5% vermicompost, Control: kitchen waste+sawdust.

The C/N ratio initially increased; however, the values recorded for the initial and final compost products were not significantly different (Figure 3b). The final C/N ratio values of all treatments ranged between 29 and 54, which were higher than the standard value (<25), indicating compost maturity. In particular, compared to the additive treatments, a more substantial increase in C/N ratio was found in the control treatment, possibly due to the substantial loss of NH$_3$-N. A proper C/N ratio controlled by the initial composting stock and additives can provide optimal porosity and suitable composting conditions, thereby promoting composting action.

The initial dissolved organic carbon (DOC) content was relatively high, and DOC sharply decreased to approximately 400 mg kg$^{-1}$ within 4 weeks in all composts (Figure 3a). The dissolved

organic nitrogen (DON) content in all treatment was considerably variable ranging during whole composting period, but trends in DON changes were similar for all composts examined (Figure 3d). The increase in DON concentration simultaneously occurred with increase in $NO_3$-N.

Changes in $NH_4^+$-N concentration can reflect nitrogen conversion during composting [32]. The $NH_4^+$-N content gradually decreased from the beginning of composting, which can be attributed to the $NH_3$ volatilization and the conversion from $NH_4^+$-N to $NO_3^-$-N (Figure 3e,f). A <40 °C temperature and aeration are favorable for nitrification [33]. High temperature in the initial stage can inhibit the activity of nitrifying bacteria [34,35]. Therefore, the $NO_3^-$-N content of the five treatments started increasing from week 5 of composting. This trend was consistent with the PCA results. $NH_4$-N showed positive correlation with $NH_3$, but negative correlation with $NO_3$-N (Figure 4).

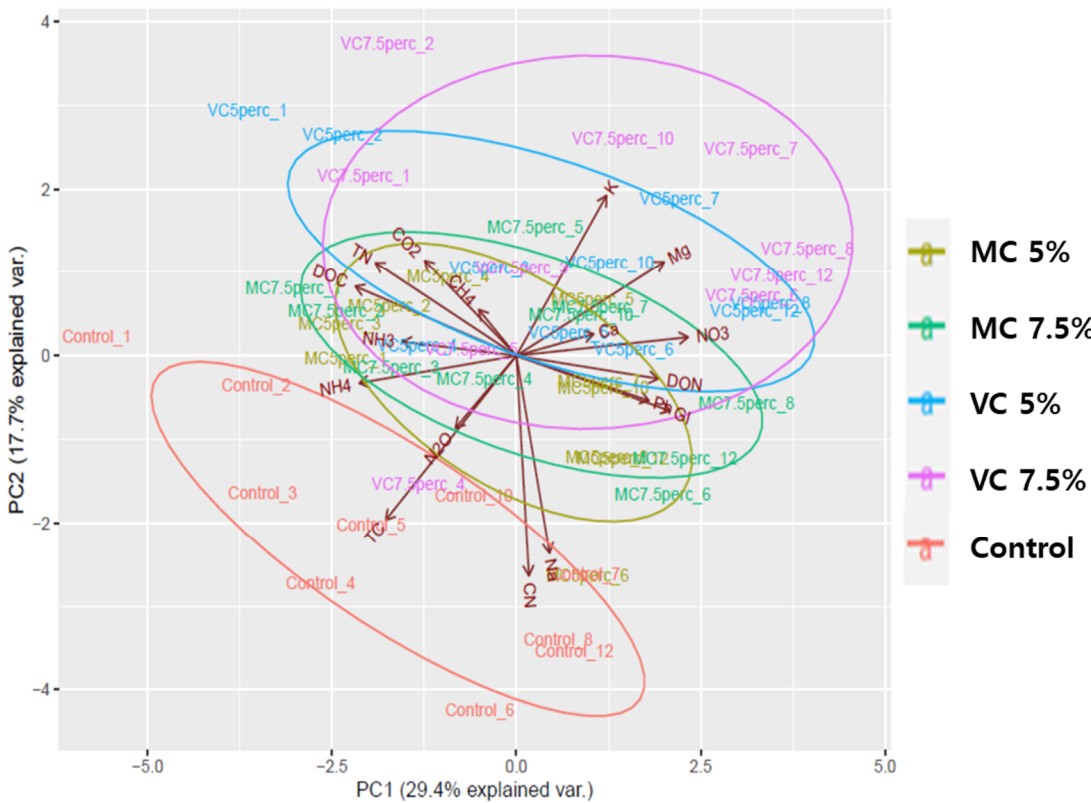

**Figure 4.** Principal-component analysis of correlation among physiochemical properties and ammonia and greenhouse gas emissions. MC 5%: kitchen waste+sawdust+5% mature compost; MC 7.5%: kitchen waste+sawdust+7.5% mature compost; VC 5%: kitchen waste+sawdust+5% vermicompost; VC 7.5%: kitchen waste+sawdust+7.5% vermicompost, Control: kitchen waste+sawdust.

When the composting process was completed, the total nitrogen losses were 12.8%, 7.7%, 24.5%, 18.5% and 60.0% for the 5% MC, 7.5% MC, 5% VC, 7.5% VC and control treatment, respectively. Hence, the addition of mature compost and vermicompost significantly decreased nitrogen loss in comparison with the control. This considerable loss of N in the control treatment led to a relatively higher C/N ratio and overall lower nutrient status (N, K, Ca and Mg) in the final compost product (Table 2).

**Table 2.** Mass and characteristics of initial and final compost.

| Treatment [1] | Composting Cycle | Mass (kg) | Content (%) | | | | | | |
|---|---|---|---|---|---|---|---|---|---|
| | | | TOC | TN | C/N | K | Ca | Mg | Na |
| MC 5% | Initial | 21.0 | 40.1 | 1.08 | 37.1 | 0.40 | 2.50 | 0.25 | 0.59 |
| | Final | 18.7 | 37.2 | 1.00 | 37.2 | 0.43 | 2.97 | 0.32 | 0.95 |
| MC 7.5% | Initial | 21.5 | 40.1 | 1.09 | 36.8 | 0.51 | 2.83 | 0.20 | 0.56 |
| | Final | 19.4 | 38.9 | 1.07 | 36.4 | 0.46 | 9.01 | 0.5 | 1.14 |
| VC 5% | Initial | 21.0 | 36.7 | 1.25 | 29.4 | 0.62 | 1.93 | 0.27 | 0.47 |
| | Final | 18.2 | 29.5 | 1.03 | 28.6 | 0.50 | 4.14 | 0.61 | 0.88 |
| VC 7.5% | Initial | 21.5 | 32.3 | 0.99 | 32.6 | 0.46 | 2.44 | 0.32 | 0.35 |
| | Final | 19.5 | 26.8 | 0.81 | 33.1 | 0.68 | 3.03 | 0.61 | 0.62 |
| Control | Initial | 20.0 | 45.2 | 1.18 | 38.3 | 0.31 | 0.72 | 0.08 | 0.66 |
| | Final | 12.3 | 42.1 | 0.78 | 54.0 | 0.39 | 1.41 | 0.20 | 1.08 |

[1] MC 5%: kitchen waste+sawdust+5% mature compost; MC 7.5%: kitchen waste+sawdust+7.5% mature compost; VC 5%: kitchen waste+sawdust+5% vermicompost; VC 7.5%: kitchen waste+sawdust+7.5% vermicompost, Control: kitchen waste+sawdust.

### 3.4. Total Gas Fluxes and Global Warming Potential

The use of two additives (mature and vermicompost) had significant effect on reducing the $NH_3$ volatilization during composting (29–69% compared to control). $NH_3$ reduction was more dependent on addition rates than additive types. The 7.5% MC and VC treatments significantly reduced $NH_3$ emission by 54–69%, which was a much greater reduction than that achieved by the 5% treatments (29–31%) (Table 3). The 7.5% VC addition was more effective in controlling $NH_3$ emissions. Proper porosity of the initial compost stock plays an important role in reducing the $NH_3$ emission [6]. Furthermore, vermicompost is widely known for its physiochemical properties, such as high porosity, good aeration, drainage and microbial activity [36].

**Table 3.** Ammonia, greenhouse gas and global warming potential (GWP) during composting.

| Compost [1] | $NH_3$ | $CO_2$ | $N_2O$ | $CH_4$ | Total GWP |
|---|---|---|---|---|---|
| | g kg$^{-1}$ D.W | | | | kg $CO_2$ eq. kg$^{-1}$ D.W |
| MC 5% | 34.8 | 1885 | 0.11 | 29.8 | 2.66 |
| MC 7.5% | 22.6 | 2556 | 0.13 | 47.9 | 3.79 |
| VC 5% | 34.0 | 4206 | 0.27 | 69.5 | 6.02 |
| VC 7.5% | 15.3 | 3758 | 0.27 | 46.4 | 5.00 |
| Control | 48.9 | 3240 | 4.73 | 88.1 | 6.85 |

[1] MC 5%: kitchen waste+sawdust+5% mature compost; MC 7.5%: kitchen waste+sawdust+7.5% mature compost; VC 5%: kitchen waste+sawdust+5% vermicompost; VC 7.5%: kitchen waste+sawdust+7.5% vermicompost, Control: kitchen waste+sawdust.

The production of $CO_2$ and $CH_4$ is mainly responsible for microbial activity; the maximal $CO_2$ emissions were observed in the 5% VC treatment, while the lowest $CO_2$ emissions were recorded in the 5% MC. Compared with the control, the mature compost treatments (5 and 7.5% MC) reduced $CO_2$ emissions by 40% and 20%, whereas the vermicompost treatments (5 and 7.5% VC) increased $CO_2$ emissions by 30% and 16%, respectively. This was likely due to the higher nutrient profile in vermicompost than that in traditional compost, as reported by many studies [37,38], and the relatively lower C/N ratio of vermicompost (Table 1).

In this study, mature compost and vermicompost reduced $CH_4$ emissions by 21–66% in comparison with the control treatment (Table 3). The additives could accommodate the increase in porosity of the compost pile, thereby mitigating the anaerobic conditions leading $CH_4$ production [39].

The principle component analysis (PCA) can express the relation between gas emissions and chemical properties during the composting process (Figure 4). A cumulative proportion of 47% contributed by the selected factors. Data distribution of control treatment was differently found, compared to additives. Gases had positive correlation with temperature and C/N ratio (Figure 4). In particular, $N_2O$ emissions showed a significantly positive correlation with total carbon, Ca and Mg content ($p < 0.05$). Total nitrogen was positively correlated with $NH_3$ emissions but negatively correlated with GI ($p < 0.05$) (Figure 5).

Total global warming impact is expressed as $CO_2$ equivalents using a global warming potential (GWP) of 1 for $CO_2$, 25 for $CH_4$, and 298 for $N_2O$ [4]. The total GWP value ranged from 2.7 to 6.9 kg $CO_2$ eq. kg$^{-1}$ dry material. The 5% MC treatment showed the greatest GWP reduction of 61% (2.7 kg $CO_2$ eq. kg$^{-1}$), while the 7.5% MC, 7.5% VC and 5% VC treatments followed with reductions of 45% (3.8 kg $CO_2$ eq. kg$^{-1}$), 27% (5.0 kg $CO_2$ eq. kg$^{-1}$) and 12% (6.0 kg $CO_2$ eq. kg$^{-1}$) compared to the control, respectively. In additive treatments, the contribution of the three GHGs to total GWP was biased, with values of 67–75% for $CO_2$, 1.0–1.6% for $N_2O$ and 23–32% for $CH_4$. The control treatment showed a relatively even contribution pattern, with values of 47% for $CO_2$, 21% for $N_2O$ and 32% for $CH_4$. The lowest values of $NH_3$ production and GWP were found in the 7.5% VC and 5% MC treatments, respectively.

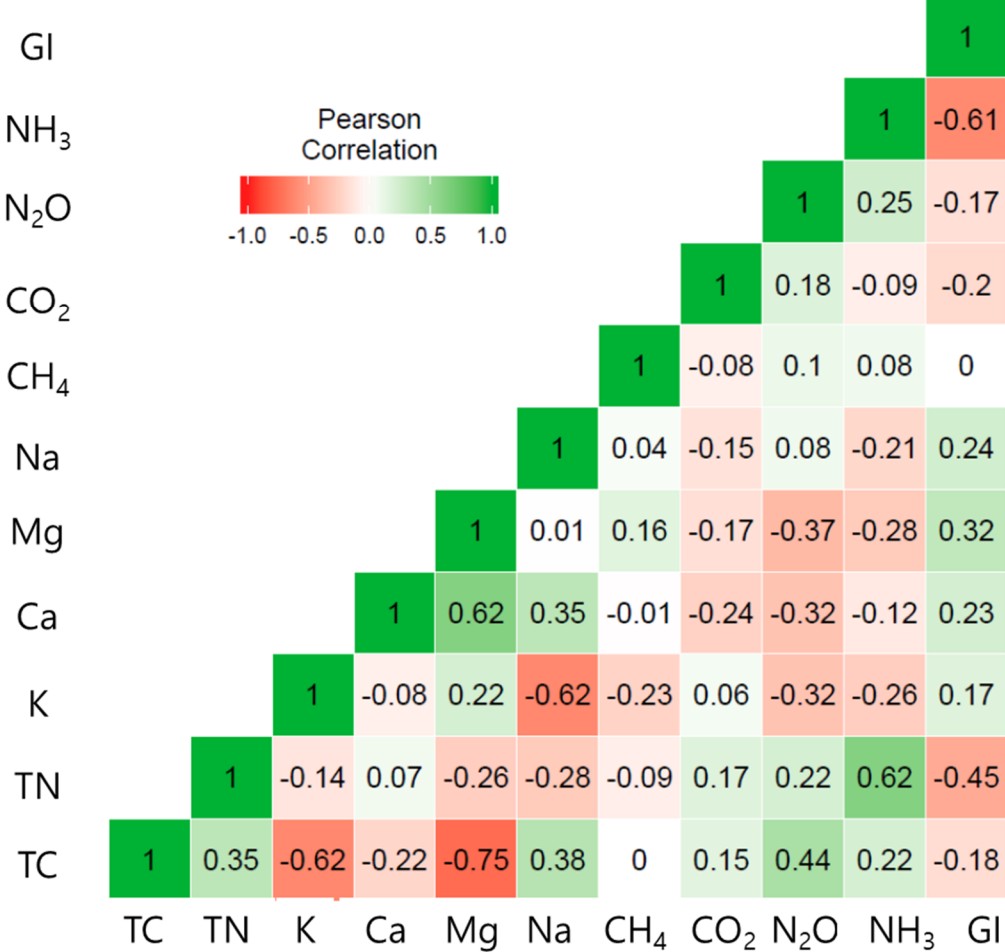

**Figure 5.** Pearson correlation analysis based on heat map of gaseous emissions and chemical properties during composting for each treatment MC 5%: kitchen waste+sawdust+5% mature compost; MC 7.5%: kitchen waste+sawdust+7.5% mature compost; VC 5%: kitchen waste+sawdust+5% vermicompost; VC 7.5%: kitchen waste+sawdust+7.5% vermicompost, Control: kitchen waste+sawdust.

## 4. Conclusions

This research indicated that the two additives reduced greenhouse gas and $NH_3$ emissions, and improved compost quality. The addition of vermicompost effectively inhibited $NH_3$ emissions, while the addition of mature compost significantly decreased global warming impact. After composting, various properties (total organic carbon (TOC), total nitrogen (TN), and C/N ratio) and the nutrient status (K, Ca, and Mg) of MC and VC treatments were better than those of the control. Although these results prove that modification with MC and VC is suitable for alleviating gas emissions and improving quality for food waste composting, a pilot-scale study is needed to assess its commercial and practical feasibility. As an additive, mature compost or vermicompost are strongly recommended for being beneficial to the composting process depending on the target gas to be reduced.

**Author Contributions:** Conceptualization, H.Y.H., S.H.K. and S.J.P.; data curation, H.Y.H., S.H.K., J.S. and S.J.P.; investigation, H.Y.H. and S.H.K.; methodology, H.Y.H., S.H.K., J.S. and S.J.P.; supervision, S.H.K., J.S. and S.J.P.; writing—original draft, H.Y.H.; writing—review and editing, H.Y.H., S.H.K. All authors have read and agreed to the published version of the manuscript.

**Funding:** This research received no external funding.

**Acknowledgments:** This research was supported by the Rural Development Administration (PJ01346801), Republic of Korea.

**Conflicts of Interest:** The authors declare no conflict of interest.

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
