# Peer review of "Composting Process and Gas Emissions during Food Waste Composting under the Effect of Different Additives"

_sustainability, doi:10.3390/su12187811_

Round 1

Reviewer 1 Report

This manuscript focused on testing the emissions of food waste composting under mature compost and vermicompost treatment. In all, the manuscript included the necessary content and was structured well. However, there are still several specific comments and suggestions for improvement, which are outlined below:

  • In the methodology session, the author should provide more explanation of the composting materials. How much the food waste was treated? What type of food waste was composted? Vegetables or meat? How much sawdust added (a mixing ratio)? 
  • There is also a concern over using vermicompost. It has to be purchased from the market. In real-life composting, it will increase the cost of the process. The ideal case is using the available wastes/feed to do composting. I wonder if manure compost treatment or vermicompost treatment would be practical in real life application.
  • More explanations should be added for Figures 4 and 5. They are difficult to understand. The characters in Figure 5 are too small and cannot tell. So cannot check these two figures.
  • What is the GWP of composting comparing to the landfill GWP?  

Author Response

I deeply appreciated all the comment and suggestions, and this manuscript was improved by MDPI English editing service (invoice ID: English-22006).

Comment 1. In the methodology session, the author should provide more explanation of the composting materials such as amount of food waste, type of food waste and mixing ratio of sawdust.

Response: I added details of composting materials in 2.1. composting materials [Food waste was obtained from enclosed, local municipal waste collection station (Damyang, Korea). The effective composition was vegetables, fruits, staple food, meat and others. The food waste (30kg wet weight) was mixed with 30% (w/w) sawdust (9kg wet weight) to control the C/N between 20 and 30 and the initial moisture content to about 60%.].

Comment 2. There is also a concern over using vermicompost. It has to be purchased from the market. In real-life composting, it will increase the cost of the process. The ideal case is using the available wastes/feed to do composting. I wonder if manure compost treatment or vermicompost treatment would be practical in real life application.

Response: I totally agreed with your comment. I considered commercial feasibility, then decided the 5-7.5% addition rate. I also added your concern in 4. conclusion part.

Comment 3. More explanations should be added for Figures 4 and 5. They are difficult to understand. The characters in Figure 5 are too small and cannot tell. So cannot check these two figures.

Response: I revised the figure 4 and 5 to clearly understand and added more description in 3. Results and discussion.

Comment 4. What is the GWP of composting comparing to the landfill GWP?

Response: Composting is a recommendable way to minimize global warming impact and cost. According to Gao et al. (2017), landfill impact on climate change is ten times greater than that of anaerobic digestion, incineration, and composting. It is not reasonable to directly compare GWP value between landfill and composting from different research cases, since gas emission were totally depended on food waste composition, duration, location, process condition, etc.

Reviewer 2 Report

The article is interesting because of the investigated issue of utilization of food waste, which is a difficult waste due to its chemical composition and structure.
In my opinion, tests should be done with a larger amount of MC or VC additive due to the possibility of determining the percentage of these additives will no longer have a significant impact on reducing harmful gas emissions. Maybe an addition of 10% maybe 15% would have better results? Without this, the study seems incomplete to me.
The conclusions are too enigmatic. They need to be developed.
Line 67 - (GHG, CO2, CH4, and N2O).
Line 73 - Exact timing for mixing and turning the compost should be given. They have a significant impact on the course of gas emissions during composting.
Line 79- EC values ​​were measured - there is no explanation what an EC is and what it was measured for. In Table 1 it is also given and not used in the article.
Line 91 - using R software at 0.05 level. - Level of what?
Line 101 - the position cited is not appropriate here due to the significantly different substrates.
Line 182 - The use of two additives (mature and vermicompost) - may be - or?
Table 2 is not included in the text.
Fig. 2 is illegible; it should start from scratch. It suggests that, for example, MC 7.5%, when inserted into a bioreactor, is less than 58 degrees Celsius. Why does it cover 11 weeks when the remaining 12?
Fig. 4 and Fig. 5. are hardly legible. 

The data provided in the tables and not commented on in the text add nothing to the article.

Author Response

I deeply appreciated all the comment and suggestions, and this manuscript was improved by MDPI English editing service (invoice ID: English-22006).

Comment 1. In my opinion, tests should be done with a larger amount of MC or VC additive due to the possibility of determining the percentage of these additives will no longer have a significant impact on reducing harmful gas emissions. Maybe an addition of 10% maybe 15% would have better results? Without this, the study seems incomplete to me.

Response: 5-10 Mg compost day-1 is averagely produced in company, then even 5 and 7.5% additives addition (350-750 kg) is not small for commercial. According to your comment, I mentioned the concerns about the research scale and commercial feasibility in 4. Conclusion part.

Comment 2. The conclusions are too enigmatic. They need to be developed.

Response: I properly revised 4. Conclusion part to suggest clear message.

Comment 3. Line 67 - (GHG, CO2, CH4, and N2O).

Response: I mentioned the abbreviation and chemical formula.

Comment 4. Line 73 - Exact timing for mixing and turning the compost should be given. They have a significant impact on the course of gas emissions during composting.

Response: I explained the timing for compost mixing in 2.3. Sampling and analytical methods [During the 84 days composting process, compost piles were turned over and thoroughly mixed every week at the fixed time of 9 am.]

Comment 4. Line 79- EC values ??were measured - there is no explanation what an EC is and what it was measured for. In Table 1 it is also given and not used in the article.

Response: Electrical conductivity (EC) is an index of the salinity of the food waste, and I added explanation. Table 1 showed the properties of composting materials, therefore no need to explain the content. Table 1 was referred to explain the highest CO2 emission from vermicompost treatment in 3.4. Total Gas Flux and Global-Warming Potential.

Comment 5. Line 91 - using R software at 0.05 level. - Level of what?

Response: 0.05 level indicated a statistical significance level, p-value, which is commonly used to hypothesis test.

Comment 6. Line 101 - the position cited is not appropriate here due to the significantly different substrates.

Response: I changed the reference concerning food waste composting similar with this study.

Comment 7. Table 2 is not included in the text.

Response: I put the explanation about compost properties according to table 2 in 3.3. Chemical characteristics of composts.

Comment 8. Fig. 2 is illegible; it should start from scratch. It suggests that, for example, MC 7.5%, when inserted into a bioreactor, is less than 58 degrees Celsius. Why does it cover 11 weeks when the remaining 12?

Response: This revised figure 2. including 12 weeks of whole composting period. Color and size of symbol were modified for easy-understanding.

Comment 9. Fig. 4 and Fig. 5. are hardly legible.

Response: I revised the figure 5 to clearly understand and added more description in 3. Results and discussion. Correlation between composting characteristics were clearly found in Figure 4, then detailed explanation was added.

Comment 10. The data provided in the tables and not commented on in the text add nothing to the article.

Response: I added discussion about all the table and figure in 3. Results and discussion.

Round 2

Reviewer 1 Report

The authors addressed comments in previous review. It is good to publish after fixing several gramma errors. 

Author Response

I really appreciated to your comment to improve this manuscript.

Reviewer 2 Report

Under Fig. 1 there is the caption "Days after composting" should be "Composting period (week)"
The authors, referring to Fig.1d, operate on days while there are weeks on the graph. In my opinion, even though 7 days is 1 week and 49 days is 7 weeks ... the units on the chart and in the description should correspond with each other.
The results of the DOC and DON measurements were not commented on in the discussion of the results, they should not be presented in both Fig. 3 and Table 1. The same applies to the presentation of the results of EC (electrical conductivity) measurements.
In my opinion, the article should only present the results of those measurements that are used in the content of the article. I still find the conclusions too laconic.

Author Response

Comment 1. Under Fig. 1 there is the caption "Days after composting" should be "Composting period (week)". The authors, referring to Fig.1d, operate on days while there are weeks on the graph. In my opinion, even though 7 days is 1 week and 49 days is 7 weeks ... the units on the chart and in the description should correspond with each other.
Response: I modified the caption of Figure 1, and changed all expression of ‘day’ into ‘week’ except description for temperature.

Comment 2. The results of the DOC and DON measurements were not commented on in the discussion of the results, they should not be presented in both Fig. 3 and Table 1. The same applies to the presentation of the results of EC (electrical conductivity) measurements. In my opinion, the article should only present the results of those measurements that are used in the content of the article.
Response: I added the results of DOC and DON measurements 3. Results and discussion. DOC and DON values of table 1 indicated properties of raw materials for composting experiment, whereas Figure 3. showed changes of DOC and DON concentration of compost during composting period.

Comment 3. I still find the conclusions too laconic.
Response: I tried to clarify the conclusion to better understand.

Round 3

Reviewer 2 Report

My comments were basically taken into account.